# Identification of *GLI1* and *KIAA0825* Variants in Two Families with Postaxial Polydactyly

**DOI:** 10.3390/genes14040869

**Published:** 2023-04-05

**Authors:** Safeer Ahmad, Muhammad Zeeshan Ali, Muhammad Muzammal, Amjad Ullah Khan, Muhammad Ikram, Mari Muurinen, Shabir Hussain, Petra Loid, Muzammil Ahmad Khan, Outi Mäkitie

**Affiliations:** 1Gomal Center of Biochemistry and Biotechnology, Gomal University, Dera Ismail Khan 29050, Pakistan; 2Research Program for Clinical and Molecular Metabolism, Faculty of Medicine, University of Helsinki, 00290 Helsinki, Finland; 3Children’s Hospital, University of Helsinki and Helsinki University Hospital, 00290 Helsinki, Finland; 4Folkhälsan Research Center, Folkhälsan Institute of Genetics, 00290 Helsinki, Finland

**Keywords:** polydactyly, *KIAA0825*, *GLI1*, Pakistani

## Abstract

Polydactyly is a rare autosomal dominant or recessive appendicular patterning defect of the hands and feet, phenotypically characterized by the duplication of digits. Postaxial polydactyly (PAP) is the most common form and includes two main types: PAP type A (PAPA) and PAP type B (PAPB). Type A involves a well-established extra digit articulated with the fifth or sixth metacarpal, while type B presents a rudimentary or poorly developed superfluous digit. Pathogenic variants in several genes have been identified in isolated and syndromic forms of polydactyly. The current study presents two Pakistani families with autosomal recessive PAPA with intra- and inter-familial phenotype variability. Whole-exome sequencing and Sanger analysis revealed a novel missense variant in *KIAA0825* (c.3572C>T: p.Pro1191Leu) in family A and a known nonsense variant in *GLI1* (c.337C>T: p.Arg113*) in family B. In silico studies of mutant KIAA0825 and GLI1 proteins revealed considerable structural and interactional modifications that suggest an abnormal function of the proteins leading to the disease phenotype. The present study broadens the mutational spectrum of *KIAA0825* and demonstrates the second case of a previously identified *GLI1* variant with variable phenotypes. These findings facilitate genetic counseling in Pakistani families with a polydactyly-related phenotype.

## 1. Introduction

Polydactyly (PD) or hyperdactyly is one of the most common congenital limb anomalies, phenotypically characterized by the duplication of digits, observed prenatally or instantly after birth [1]. It can be inherited as an isolated limb abnormality or as a syndromic feature. Polydactyly mainly results from inaccurate patterning along the anterior to posterior axis of the developing limb [2]. Non-syndromic polydactyly has been classified into three main types based on the position of the superfluous digit [3]. The presence of an extra digit outside the little finger and/or toe is referred to as postaxial polydactyly (PAP). The presence of an extra digit outside the thumb and/or big toe is termed preaxial polydactyly (PPD). Mesoaxial or central polydactyly is defined as the duplication of the index, long, or ring finger. Mesoaxial polydactyly is very rare and is typically associated with complex syndactyly or cleft hand [4].

PAP is categorized into two forms that include type A (PAPA) and type B (PAPB) [5]. PAPA involves a completely developed supernumerary digit articulated with either the fifth or the sixth metacarpal, while PAPB involves a rudimentary, non-functional extra digit or poorly developed soft digit with or without a defective bony element [3]. A Pakistan-based clinical and descriptive genetic study reported that PAP accounts for 52% of non-syndromic polydactyly [6]. In the literature, both syndromic and non-syndromic PAPs have been reported, segregating as an autosomal recessive or autosomal dominant trait [7].

To date, pathogenic variants in 12 genes (*GLI3, GLI1, ZNF141, IQCE, KIA0825, FAM92A1, DACH1, SMO, STKLD1, PITX1, MIPOL1* and *LMBR1*) and five loci (13q21-32, 13q13.3-21.2, 7q21-q34, 19p13.1-13.2, and 2q31.1-31.2) have been associated with isolated polydactyly [4].

The molecular processes involved in limb development are complex and still not fully understood in vertebrates and invertebrates [8]. Several cellular pathways, including Sonic Hedgehog (SHH), Transforming Growth Factor Β (TGFβ), Fibroblast Growth Factor (FGF), Wingless (WNT), Bone Morphogenetic Protein (BMP), and Notch regulate the limb development in humans [2]. A deficiency or defect in any of the regulators in these pathways can result in several limb anomalies [5,9]. The molecular etiology of polydactyly has been associated with genes involved in anterior-posterior patterning. The most important pathway in regulating anterior-posterior patterning is the SHH-GLI pathway [10].

The SHH-GLI signaling pathway is a highly conserved signaling pathway that controls cell specification, cell–cell interaction, and tissue patterning during embryonic development [11]. Components of the SHH pathway are primarily located within the cilia, which coordinate and control signaling/trafficking events. Advances in mouse genetics and genome-wide CRISPR-based research have explained the connection between SHH and cilia signaling. SHH signaling is mostly generalized within the cilia’s specialized domains, such as the transition zone that regulates cilia receptor entry, the tip region that regulates the GLI protein, and the EVC zone that scaffolds SMO signaling [4].

The role of *GLI* in limb development is well-explained in the literature [11]. *GLI3* is primarily responsible for transcription repression, while *GLI2* is responsible for transcription activation. GLI3 and GLI2 bind to GLI1 and act as transcriptional activators, increasing the overall HH activity [12]. So far, the function of *KIAA0825* has not been characterized. Some studies have been performed on the *KIAA0825* mouse orthologous gene 2210408I21Rik, which shows its expression in the developing limbs and forelimb buds. A homozygous knock-out mouse (2210408I21Riktm1b (EUCOMM) Wtsi) demonstrated several skeletal irregularities affecting growth, body size, and bone mineral density [13].

In this study, we present two unrelated consanguineous Pakistani families presenting with autosomal recessive PAPA with intra-familial phenotypic variations. Whole exome sequencing (WES), followed by Sanger sequencing, revealed a novel missense variant in *KIAA0825* (c.3572C>T: p.Pro1191Leu) in family A. In silico analysis of the variant demonstrated major structural and interactional changes in the mutant KIAA0825 protein. WES and Sanger analyses of family B revealed a nonsense variant in *GLI1* (c.337C>T: p.Arg113*). This is the second Pakistani family with this nonsense *GLI1* variant with variable phenotypes. In silico analysis showed significant structural and interactional changes in the mutant GLI1 protein.

## 2. Materials and Methods

### 2.1. Study Approval and Subjects

The institutional review boards (IRBs) of Gomal University, Pakistan, approved the current genetic study. Written informed consent was obtained from the study participants and guardians of the family for the research experiments, molecular analysis, and publication of the data. In this study, two unrelated families from different regions of Pakistan were recruited. Family A was recruited from Dera Ismail Khan, a district of Khyber Pakhtunkhwa, and family B from Bhakkhar, a district of Punjab. Both families presented non-syndromic PAPA. Detailed family histories and pedigree analyses revealed an autosomal recessive inheritance pattern of the disease in both families. Clinical examinations of all the patients were carried out in the local government hospital. Blood samples from all participants were collected, and genomic DNA extraction was performed by standard laboratory protocol.

### 2.2. Whole-Exome Sequencing and Data Analysis

Genomic DNA samples of one affected member from each family underwent whole-exome sequencing. An Agilent SureSelect Exome V6 Capture Library kit was used to prepare the exome library. Barcoded libraries were pooled, and sequencing was performed on an Illumina HiSeq with an average on-target coverage of 87×. Read alignment with human assembly hg19 (GRCh37) was done via Burrows-Wheeler Aligner (BWA v0.7.5), and variant calling was performed by different tools, including Samtools (v0.1.18), Picard (v2.26.10), and GATK (v4.2.0.0). The VCF file of SNVs and small indels was annotated by an offline version of ANNOVAR (July 2017). The variants were filtered with a minor allele frequency (MAF) ≥0.005 in gnomAD, a CADD_PHRED score of ≥20, and a Kaviar allele count of ≤10. The list of variants was further restricted to the genes in OMIM, which are associated with polydactyly. Since there was a clear autosomal recessive pattern of inheritance by pedigree analysis, we focused on rare homozygous and compound heterozygous variants.

### 2.3. Pathogenicity Index and Protein Stability

The biological pathogenicity of the variants was determined using several online bioinformatics tools, such as MutationTaster2021 https://www.genecascade.org/MutationTaster2021/ (accessed on 1 March 2023) [14], Combined Annotation Dependent Depletion (CADD) https://cadd.gs.washington.edu/ (accessed on 1 March 2023) [15], Polymorphism Phenotyping V2 (PolyPhen-2) http://genetics.bwh.harvard.edu/pph2/ (accessed on 1 March 2023) [16], Sorting Intolerant From Tolerant (SIFT) http://sift.bii.a-star.edu.sg/ (accessed on 1 March 2023) [17], fathmm http://fathmm.biocompute.org.uk/ (accessed on 1 March 2023) [18], and some others. The MAF of the variants was determined using gnomAD, ExAC and 1000 genomes, and dbSNP. The evolutionary rate of an amino (or nucleic) acid position is strongly dependent on its structural and functional importance. Clustal Omega https://www.ebi.ac.uk/Tools/msa/clustalo/ (accessed on 1 March 2023) [19] was used for the conservational analysis of the variants. As web-based tools, I-Mutant3.0 http://gpcr2.biocomp.unibo.it/cgi/predictors/I-Mutant3.0/I-Mutant3.0.cgi (accessed on 1 March 2023) [20] and MUpro http://mupro.proteomics.ics.uci.edu (accessed on 1 March 2023) [21] were used to predict the stability of the mutant KIAA0825 protein.

### 2.4. Sanger Sequencing

Sanger DNA sequencing was used to determine the segregation of the *KIAA0825* and *GLI1* variants in the affected families. Primer3web (version: 4.1.0) https://primer3.ut.ee/ (accessed on 15 August 2022) was used to design the primers [22]. *KIAA0825* and *GLI1* were bidirectionally sequenced using the di-deoxy chain termination method. Sequence analysis was done with UGENE (version 41.0) and BioEdit (version 7.0.5) software [23,24].

### 2.5. Modeling and Interaction Studies

Wildtype and mutant 3-D models of KIAA0825 and GLI1 were obtained using I-TASSER https://zhanggroup.org/I-TASSER/ (accessed on 10 December 2022) [25], and the models with the highest confidence score (C-score) were selected for further interaction analysis. STRING https://string-db.org/ (accessed on 10 December 2022) [26] was used to predict the close interactors of KIAA0825 and GLI1. The docking analysis of wildtype and mutant KIAA0825 and GLI1 proteins with their close interacting proteins was performed using ClusPro 2.0 https://cluspro.bu.edu/login.php (accessed on 18 December 2022) [27]. Chimera 1.13.1 [28] and LigPlot+ (Version 2.1) [29] were used for molecular visualization.

## 3. Results

### 3.1. Clinical Description

Family A includes two affected members (IV-2 and IV-3) with PAP, syndactyly, and clinodactyly in a recessive pattern (Figure 1A). Family A demonstrated intra-familial phenotypic differences. The affected individual IV-2 presented PAPA in both hands and feet, fused second and third digits in both feet and clinodactyly in both hands (Figure 1B) (Table 1). The other affected individual, IV-3 (not shown in the figure), had only PAPA in both hands. Family B also comprises two affected members (IV-2 and IV-5) presenting with autosomal recessive PAPA with an intra-familial phenotypic difference (Figure 1C). The affected member IV-2 presented PAPA in the left hand and feet, clinodactyly of the sixth finger, 5/6 cutaneous syndactyly in the right foot, a sandal gap, and an abnormal shape of the big toe (Figure 1D). The affected individual IV-5 demonstrated a bilateral PAPA in the hands (Figure 1D) and had normal feet (Table 1). No facial dysmorphism, dental abnormalities, hearing problems, or heart defects were observed in any of the family members. Developmental delays were not observed, and all of the affected members from each family had a normal skull, nails, eyesight, weight, and height.

### 3.2. Molecular Findings

DNA samples from affected individuals IV-2 (family A) and IV-2 (family B) were subjected to whole-exome sequencing. Variant filtering in family A revealed a novel missense variant (c.3572C>T: p.Pro1191Leu) in exon 19 of the *KIAA0825* gene (Table 1 and Table 2). According to ACMG classification, the identified *KIAA0825* variant is classified as a variant of uncertain significance. In gnomAD, there are nine heterozygotes of this variant found in the South Asian region. However, the variant is absent in South Asian population databases like GenomeAsia100k, IndiGenomes, and South Asian Genome and Exome (SAGE). The segregation analysis of family A confirmed the co-segregation of the missense *KIAA0825* variant with the disease phenotype. The mutation analysis demonstrated IV-2 as being mutation homozygous affected and IV-4 as wildtype homozygous unaffected in family A (Figure 2A). The mutation was novel and had not previously been described.

In family B, a variant analysis revealed a previously reported nonsense variant (c.337C>T: p.Arg113*) in exon 4 of the *GLI1* gene (Table 1 and Table 2) [30]. The Sanger analysis confirmed the co-segregation of the nonsense *GLI1* variant with the disease phenotype. The mutation analysis demonstrated IV-2 and IV-5 as mutation homozygous affected and IV-4 as wildtype homozygous unaffected (Figure 3A).

### 3.3. Pathogenicity Validation and Protein Stability

The predicted biological pathogenicity of the identified variants is presented in Table 2. Multiple sequence alignments through Clustal Omega revealed high conservation of Pro1191 (KIAA0825) and Arg113 (GLI1) across several species (Figure 4). I-Mutant3.0. predicted the decreased stability of the mutant (p.Pro1191Leu) KIAA0825 protein with a reliability index of 2 and a DGG value prediction of 0.16Kcal/mol. Mupro also predicted the mutant KIAA0825 protein with decreased stability, having a confidence score of −0.5.

### 3.4. Modeling and Docking Analysis

Three-dimensional structures of the wildtype and mutant KIAA0825 were modeled by I-TASSER (Figure 5A,B). By structure comparison, the wildtype and mutant KIAA0825 showed 0.71 percent identity and failed to completely superimpose on each other (Figure 5C). The docking analysis revealed a significant interaction alteration between the wildtype and mutant KIAA0825 proteins with the close interactor C1orf16. Four residues of the wildtype KIAA0825 made an interaction with five residues of C1orf167 via three hydrogen bonds and two salt bridges, while eight residues of the mutant KIAA0825 interacted with nine residues of C1orf167 via seven hydrogen bonds and two salt bridges (Figure 6A,B) (Appendix A).

The wildtype and mutant 3-D models of GLI1 were also generated by I-TASSER (Figure 7A,B). The wildtype and mutant GLI1 failed to superimpose on each other (Figure 7C) and showed 9.82 percent identity. Interactions of the wildtype and mutant GLI1 proteins with close interactor (SUFU) demonstrated interactional changes. Five residues of the wildtype GLI1 interacted with six residues of SUFU via three hydrogen bonds and three salt bridges, while four residues of each mutant GLI1 and SUFU interacted via four hydrogen bonds (Figure 8A,B) (Appendix A).

## 4. Discussion

The present study reports two unrelated consanguineous Pakistani families with autosomal recessive PAPA with intra- and inter-familial phenotypic variations. Family A included two affected members who inherited PAPA with a recessive inheritance pattern. The individual IV-2 of family A presented PAPA in both hands and feet, cutaneous syndactyly in both feet, and clinodactyly in both hands, while individual IV-3 of family A had PAPA in both hands but no other hand or feet abnormalities. WES of family A revealed a novel missense variant (c.3572C>T: p.Pro1191Leu) in the *KIAA0825* gene. The identified variant co-segregated with the disease phenotype. The *KIAA0825* gene is mapped on chromosome 5q15 and encodes a 553 amino acids protein. So far, the function of the KIAA0825 protein has not been characterized. A few studies have been performed on the mouse orthologous gene *2210408I21Rik.* The expression of *2210408I21Rik* was observed in the developing limb buds from E11.5 to E15, and a homozygous knock-out mouse (*2210408I21Rik^tm1b(EUCOMM)Wtsi^*) demonstrated several skeletal irregularities affecting growth, body size, and reduction in bone mineral density [13]. To date, a total of six other pathogenic variants in *KIAA0825* have been reported in subjects with PAP. Four pathogenic variants [p.(Gln198Thrfs*21), p.(Lys725*), p.(Leu17Ser), and p.(Cys48Serfs*28)] were reported in Pakistani families [7,8,31] as well as two splice site variants [c.-1-2A>T and c.2247-2A>G] in a Chinese fetus (Table 3) [32].

A phenotypic comparison between the present study and the previous reports of *KIAA0825* is summarized in Table 4. KIAA0825 consists of a long isoform that includes 1275 amino acids and a small isoform that includes 324 amino acids. The variant identified in our study affects only the long isoform.

Family B is the second Pakistani family with PAP, having a previously reported nonsense variant (c.337C>T: p.Arg113*) in the *GLI1* gene. The WES of family B revealed a known nonsense variant (c.337C>T: p.Arg113*) in the *GLI1* gene, and the identified variant was co-segregated with the disease phenotype. We are unable to assess the segregation of the variant in the parents. In the presence of two homozygous siblings in the family, parents can be considered obligate carriers of the variant. GLI1 acts as a pathway mediator for hedgehog signaling. When the hedgehog molecule binds to its receptor, the GLI proteins become active, resulting in the transcription of target genes involved in bone formation and patterning. *GLI1* is mapped on chromosome 12q13.3 and encodes an 1106 amino acid protein [33]. Several domains are present in the GLI1 protein, including degron degradation signals (amino acids Dn 77–116 and Dc 464–469), SUFU binding domains (amino acids SU 111–125 and C-terminus), a zinc finger domain (ZF amino acids 235–387), nuclear localization signal (NLS 380–420 amino acids), and the transactivation domain (1020–1091 amino acid) [33]. Previously, Palencia-Campos et al. (2017) reported the same stop gain *GLI1* variant (p.Arg113*) in a Pakistani family with non-syndromic PAPA [30]. Their affected individuals had presented bilateral PAPA in feet, and one individual (patient 7) with additional PAP in the left hand. The affected member (IV-2) of the present Family B displayed the same phenotype as patient 7 of the Palencia report. However, additionally, clinodactyly of the sixth finger and 5/6 cutaneous syndactyly in the right foot were also observed in the same individual (IV-2) of the present study. Individual IV-5 presented PAPA only in the hands, while the feet were normal. This shows a considerable phenotypic difference in the present study in comparison to the previous findings by Palencia. So far, four Pakistani and two Turkish families showing intra-and inter-familial phenotypic variations in polydactyly due to mutation in *GLI1* (Table 5) have been reported. A phenotypic comparison between the present study and the previous reports of *GLI1* is summarized in Table 6. The identified variant p.(Arg113*) in family B, located in the degron degradation signals domain, results in the truncation of the protein before the Zinc finger domains. Previously, two variants, p. (Thr355Asn) and p.(Ser378Leu), were reported in the ZF motif region, and three others [p.(Leu506Gln); p.(Gln644*); p.(Trp780*)] were mapped between the transactivation domain and the nuclear localization signal.

## 5. Conclusions

We present two unrelated Pakistani families having PAPA with intra- and inter-familial phenotypic variability. These two different families harbor the two different genetic variants in *KIAA0825* and *GLI1* underlying the disease. We have identified the seventh novel variant in *KIAA0825* causing polysyndactyly and clinodactyly. The novel variant in *KIAA0825* expands the mutational spectrum and demonstrates new molecular signaling cascades required for proper limb orientation and growth. A known variant of *GLI1* was found in a second Pakistani family in a different region with phenotypic variability with respect to previous findings. As a transcription factor, the presence of a truncated/abnormal GLI1 protein can disrupt the function of the downstream genes responsible for limb development. The present study will also aid in genetic counseling of affected families with polydactyly.

## Figures and Tables

**Figure 1 genes-14-00869-f001:**
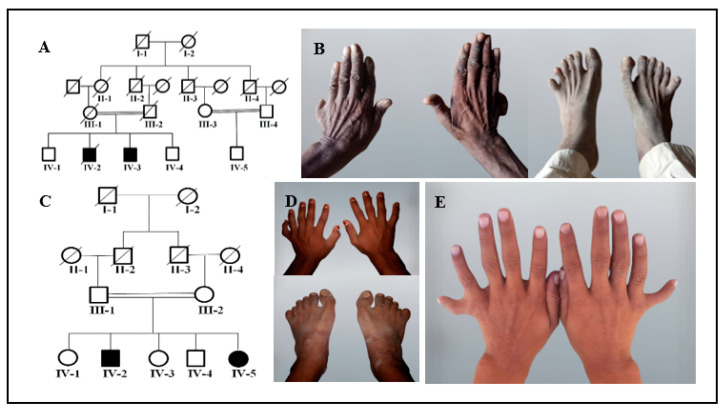
Pedigrees and images of family A and family B. (**A**) Pedigree of family A showing autosomal recessive inheritance. Blank “Square” and “circle” presents normal males and females while shaded “square” and “circle” presents affected males and females, respectively. A diagonal line over the “square” and “circle” indicates deceased individuals. Double lines between “square” and “circle” shows the consanguineous marriage. (**B**) Images of affected member IV-2 of family A displaying PAPA in both hands and feet, 2/3 cutaneous syndactyly in both feet and clinodactyly in both hands. (**C**) Pedigree of family B presenting autosomal recessive inheritance. Blank “Square” and “circle” presents normal males and females while shaded “square” and “circle” presents affected males and females, respectively. A diagonal line over the “square” and “circle” indicates deceased individuals. Double lines between “square” and “circle” shows the consanguineous marriage. (**D**) Images of the hands and feet of individual IV-2 of family B showing PAPA in the left hand and both feet, sixth finger clinodactyly in the right hand, 5/6 cutaneous syndactyly in the right foot, sandal gap, and abnormal big toe shape. (**E**) Image of individual IV-5 from family B demonstrating bilateral PAPA in hands.

**Figure 2 genes-14-00869-f002:**
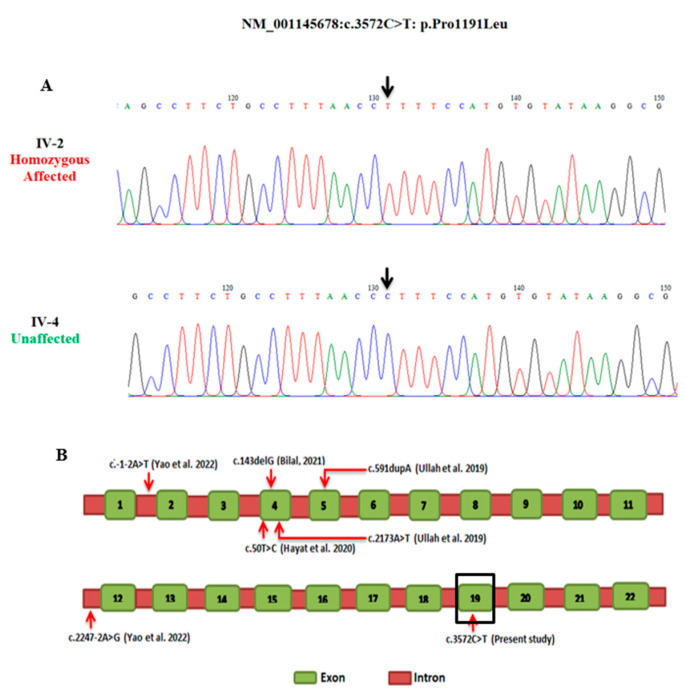
Sequence chromatogram of exon 19 of *KIAA0825*. Panel (**A**) presents individual IV-2 as homozygous affected and IV-4 as homozygous wild type. Arrow highlights the position of the substitution mutation (c.3572C>T: p.Pro1191Leu). Panel (**B**) presents a schematic diagram of the *KIAA0825* gene structure showing exons. Red arrows indicate the position of mutations reported in polydactyly to date. Variant found in the present study is marked in black square.

**Figure 3 genes-14-00869-f003:**
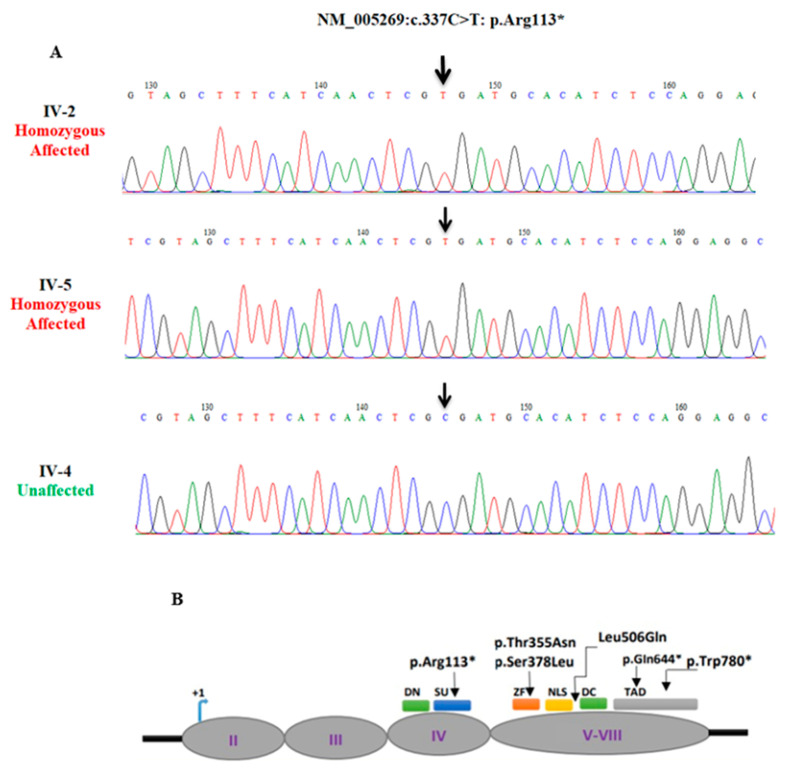
Sequence chromatogram of exon 4 of *GLI1*. Panel (**A**) presents individuals IV-2 and IV-5 as homozygous affected and IV-4 as homozygous wild type. Arrow highlights the position of the substitution mutation (c.337C>T: p.Arg113*). Panel (**B**) presents the coding areas of GLI1 and anticipated domains, and the arrows indicate the reported mutations in polydactyly to date. (DN/DC: degron degradation signals, SU: SUFU binding domains, ZN: a zinc finger domain, NLS: nuclear localization signal, TAD: transactivation domain).

**Figure 4 genes-14-00869-f004:**
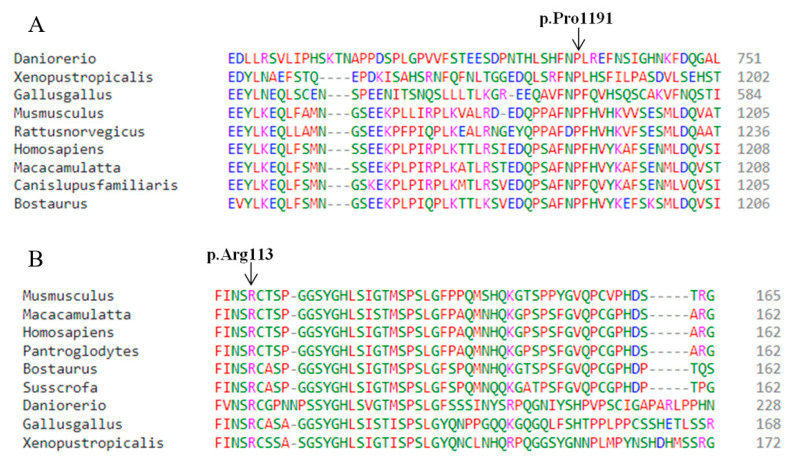
Conservation profile of KIAA0825 and GLI1 across several species predicted by Clustal Omega. (**A**) Conservation profile of Pro1191 (KIAA0825). (**B**) Conservation profile of Arg113 (GLI1).

**Figure 5 genes-14-00869-f005:**
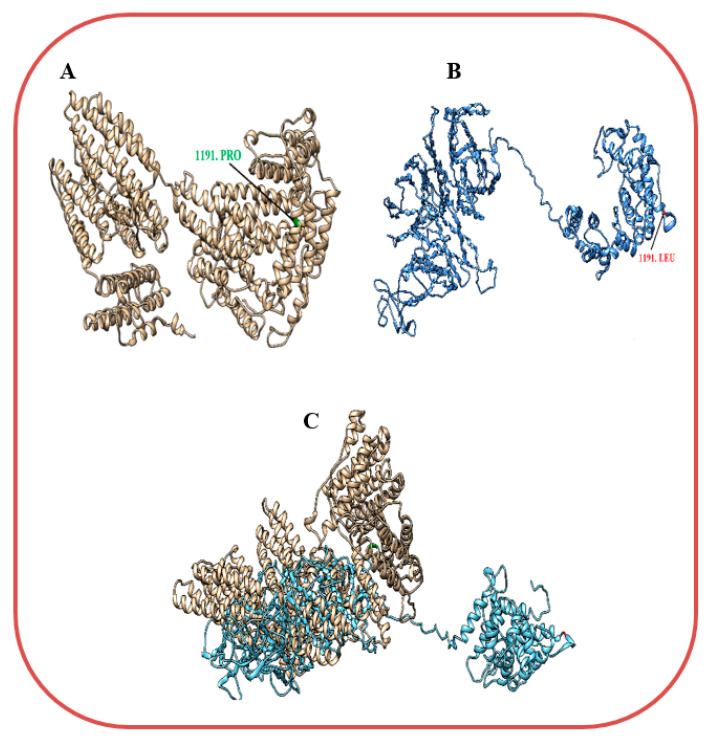
Three-dimensional structural presentations of KIAA0825 predicted by I-Tasser (**A**) Wildtype KIAA0825 (**B**) Mutant KIAA0825 (**C**) Superimposed Wildtype and mutant KIAA0825.

**Figure 6 genes-14-00869-f006:**
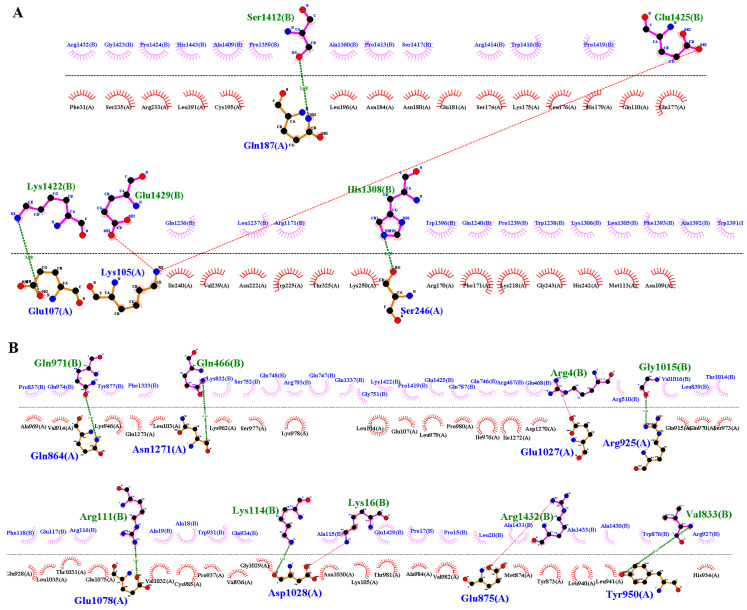
Interactions of wildtype and mutant KIAA0825 with close interactor C1orf167. (**A**) Four residues (chain A) of the wildtype KIAA0825 interacting with five residues (chain B) of C1orf167 via three hydrogen bonds and two salt bridges. (**B**) Eight residues (chain A) of the mutant KIAA0825 interacting with nine residues (chain B) of C1orf167 via 7 hydrogen bonds and two salt bridges.

**Figure 7 genes-14-00869-f007:**
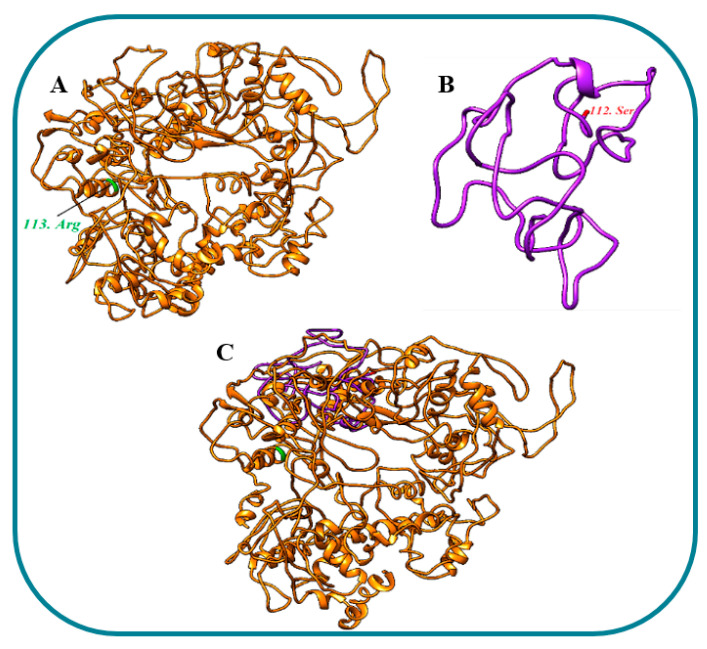
Three-dimensional structural presentations of GLI1 predicted by I-TASSER (**A**) Wildtype GLI1 (**B**) Mutant KIAA0825 (**C**) Superimposed wildtype and mutant GLI1.

**Figure 8 genes-14-00869-f008:**
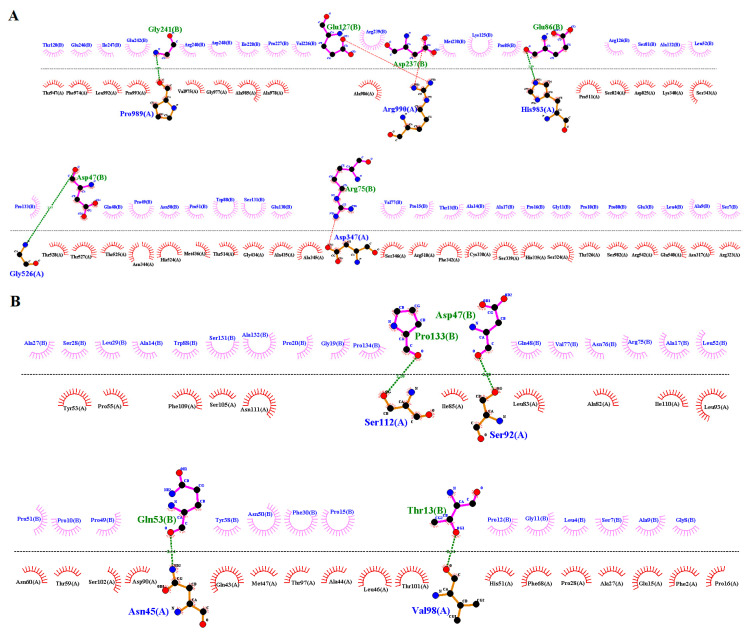
Interactions of the wildtype and mutant GLI1 with their close interactor SUFU. (**A**) Five residues (chain A) of wild type GLI1 interacting with six residues (chain B) of SUFU via three hydrogen bonds and three salt bridges. (**B**) Four residues (chain A) of mutant GLI1 interacting with four residues (chain B) of SUFU via four hydrogen bonds.

**Table 1 genes-14-00869-t001:** Clinical presentation of the affected members in two unrelated families.

Family	Clinical Examinations						Gene/ Mutation
	Subject	Polydactyly	Syndactyly	Clinodactyly	Brachydactyly	Other Anomaly	
A	IV-2 (Male)	Bilateral PAPA of hands and bilateral PAPA of feet	Bilateral 2/3 cutaneous syndactyly of feet	Bilateral clinodactyly of hands	−	Sandal gap abnormal big toe shape	*KIAA0825* c.3572C>T p.Pro1191Leu
	IV-3 (Male)	Bilateral PAPA of hands	−	−	−	−	
B	IV-2 (Male)	Unilateral PAPA of hand and bilateral PAPA of feet	−	−	−	−	*GLI1* c.337C>T p.Arg113*
	IV-5 (Female)	Bilateral PAPA of hands	−	−	−	−	

**Table 2 genes-14-00869-t002:** Detail of the variants and in silico pathogenicity predictions.

Gene		Zygosity	Genomic Position (hg19)	mRNA Transcript	cDNA Change	Amino Acid Change	gnomAD Allele Count	gnomAD All	gnomAD South Asian
*KIAA0825*		Homozygous	5:93,721,994	NM_001145678	c.3572C>T	p.Pro1191Leu	9 Heterozygote	0.00005731	0.0003953
	Tools	CADD_phred	Mutation Taster	Mutation Assessor	M-cap	Sift	Polyphen-2	FATHMM	Provean
	Predictions	33	Disease causing	Medium	Damaging	Damaging	Damaging	Damaging	Damaging
*GLI1*		Zygosity	Genomic position (hg19)	mRNA Transcript	cDNA change	Amino acid change	gnomAD allele count	gnomAD All	gnomAD South Asian
		Homozygous	12:57,858,599	NM_005269	c.337C>T	p.Arg113*	1 Heterozygote	0.00000397	0.0000326
	Tools	CADD_phred	Mutation Taster	FATHMM					
	Predictions	36	Disease causing	Damaging					

**Table 3 genes-14-00869-t003:** Description of previously reported mutations in *KIAA0825* in association with PAP.

Study	Phenotype	Mutation Type	Nature	cDNA	Amino Acid Change	Ethnicity
Yao et al., 2022 [32]	PAPA	Splice site	HET	c.-1-2A>T	−	Chinese
Yao et al., 2022 [32]	PAPA	Splice site	HET	c.2247-2A>G	−	Chinese
Bilal and Ahmad, 2021 [8]	PAPA	Frameshift	HZ	c.143delG	p. 28Cys>Ser48fs*	Pakistani
Hayat el at., 2020 [7]	PAPA	Missense	HZ	c.50T>C	p. 17Leu>Ser	Pakistani
Ullah et al., 2019 [31]	PAPA/B	Frameshift	HZ	c.591dupA	p. 21Gln>Thr198fs*	Pakistani
Ullah et al., 2019 [31]	PAPA/B	Nonsense	HZ	c. 2173A>T	p. Lys 725*	Pakistani
Present study	PAPA with cutaneous syndactyly	Missesnse	HZ	c. 3572 C >T	p.1191Pro >Leu	Pakistani

PAP: Postaxial polydactyly, HZ: Homozygous, HET: Heterozygous, *: Terminated.

**Table 4 genes-14-00869-t004:** Phenotypic comparison of the patients in the present study with previously reported patients with pathogenic variants in *KIAA0825*.

Study	Subject	Phenotypes									
		PAPA		PAPB		Syndactyly		Camptodactyly		Clinodactyly	
		Hands	Feet	Hands	Feet	Hands	Feet	Hands	Feet	Hands	Feet
Present study	IV-2	++	++	−	−	−	++	−	−	++	−
	IV-3	+	−	−	−	−	−	−	−	−	−
Yao et al., 2022 [32]	Fetus II-2	++	++	−	−	−	−	−	−	−	−
Bilal, 2021 [8]	IV-2	++	++	−	−	−	++	−	−	−	−
	IV-3	+	++	−	−	−	−	−	−	−	−
Hayat et at., 2020 [7]	IV-1	++	++	−	−	−	−	−	++	−	−
Ullah et al., 2019 [31]	V-1	++	++	−	−	−	−	−	−	++	−
	V-2	+	++	+	−	−	−	−	−	++	−

++: bilateral, +: unilateral, −: Absent, PAPA: Postaxial polydactyly type A, PAPB: Postaxial polydactyly type B.

**Table 5 genes-14-00869-t005:** Description of previously reported pathogenic *GLI1* variants associated with PAP.

Study	Phenotype	Mutation Type	Nature	cDNA	Amino Acid Change	Ethnicity
Bakar et al., 2022 [33]	PAPA/B	Missense	HET	c.1133C>T	p.378Ser>Leu	Pakistani
Yousaf et al., 2020 [34]	PAPA/B	Missense	HET	c.1064C>A	p.355Thr>Asn	Pakistani
Ullah et al., 2019 [35]	PPD	Missense	HZ	c.1517T>A	p.506Leu>Gln	Pakistani
Palencia et al., 2017 [30]	PAPA	Nonsense	HZ	c.337C>T	p.Arg113*	Pakistani
Palencia et al., 2017 [30]	Ranging from simple PAP to EVC syndrome	Nonsense	HZ	c.2340G>A	p.Trp780*	Turkish
Palencia et al., 2017 [30]	Ranging from simple PAP to EVC syndrome	Nonsense	HZ	c.1930C>T	p.Gln644*	Turkish
Present study	PAPA	Nonsense	HZ	c.337C>T	p.Arg113*	Pakistani

PAP: Postaxial polydactyly, PPD: Preaxial polydactyly, HZ: Homozygous, HET: Heterozygous, *: Terminated.

**Table 6 genes-14-00869-t006:** Phenotypic comparison of the patients in the present study with previously reported patients with pathogenic variants in *GLI1*.

Study	Member	Phenotypes											
		PPD		PAPA		PAPB		Syndactyly		Camptodactyly		Clinodactyly	
		Hands	Feet	Hands	Feet	Hands	Feet	Hands	Feet	Hands	Feet	Hands	Feet
Present study	IV-2	−	−	+	++	−	−	−	+	−	−	+	−
	IV-5	−	−	++	−	−	−	−	−	−	−	−	−
Bakar et al., 2022 [33]	III-2	−	−	++	++	−	−	−	−	−	−	−	−
	III-9	−	−	−	−	++	−	−	−	−	−	−	−
	IV-3	−	−	++	++	−	−	−	−	−	−	−	−
	IV-4	−	−	++	++	−	−	−	−	−	−	−	−
	IV-14	−	−	−	++	−	−	−	−	−	−	−	−
Yousaf et al., 2020 [34]	III-4	−	−	+	−	−	−	−	−	−	−	−	−
	IV-2	−	−	−	++	−	−	−	++	−	−	−	−
	IV-3	−	−	−	−	+	−	−	−	−	−	−	−
	IV-6	−	−	+	+	−	−	−	−	−	−	−	−
	IV-8	−	−	−	−	+	−	−	+	−	−	−	−
Ullah et al., 2019 [35]	VI-4	+	−	−	−	−	−	−	−	−	−	−	−
	VI-5	++	−	−	−	−	−	−	−	−	−	−	−
Palencia et al., 2017 [30]	Patient 1With EVC	−	−	++	++	−	−	−	−	−	−	−	−
	Patient 2	−	−	++	−	−	−	−	−	−	−	−	−
	Patient 3	−	−	++	−	−	−	−	−	−	−	−	−
	Patient 4	No confirmed phenotype											
	Patient 5 With EVC	−	−	++	++	−	−	−	−	−	−	−	−
	Patient 6 With EVC	−	−	++	++	−	−	−	−	−	−	−	−
	Patient 7	−	−	+	++	−	−	−	−	−	−	−	−
	Patient 8	−	−	−	++	−	−	−	−	−	−	−	−
	Patient 9	−	−	−	++	−	−	−	−	−	−	−	−
	Patient 10	−	−	−	++	−	−	−	−	−	−	−	−

++: bilateral, +: unilateral, −: Absent, PPD: Preaxial polydactyly, PAPA: Postaxial polydactyly type A, PAPB: Postaxial polydactyly type B, EVC: Ellis-van Creveld syndrome.

## Data Availability

The data presented in this study are available on request from the corresponding author. The data are not publicly available due to privacy.

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
