# Peer review of "Identification of GLI1 and KIAA0825 Variants in Two Families with Postaxial Polydactyly"

_genes, 2023, doi:10.3390/genes14040869_

Round 1
Reviewer 1 Report
Identification of GLI1 and KIAA0825 variants in two families with postaxial polydactyly
Ahmad et al. has presented clinical and genetic findings of two unrelated Pakistani families affected by postaxial polydactyly. In this they identified a novel missense variant in KIAA0825 gene and found only in the South Asian population while already reported nonsense GLI variant. They have performed in-silico analysis. This will help in improving the spectrum of the genetic and clinical characteristics of the disease. It could be helpful to identify more patients with the same variant in the disease cohort who belong to South Asian ancestry.
The manuscript is well written. However there are minor changes that author should address:
Major
-
The novel variant (3572C>T KIAA0825) was earlier submitted by a Korean company. https://clinvarminer.genetics.utah.edu/submissions-by-variant/NM_001145678.3%28KIAA0825%29%3Ac.3572C%3ET%20%28p.Pro1191Leu%29. Based on their ACMG-AMP classification they have classified this variant as VUS. Can you please perform American College of Medical Genetics and Genomics - Association of Molecular Pathology (AMCG-AMP) guidelines for both of your variants?
-
This is very interesting as the mutation in ​​KIAA0825 is only present in the gnomAD South Asian population. Can you please search for the variant in the South Asian population databases like GenomeAsia100k, IndiGenomes, and South Asian Genome and Exome (SAGE) and report it?
-
It is important to perform the Sanger sequencing of the parents to identify them as heterozygous for the disease which is inherited by the affected patients in both the families.
Minor:
-
Line 85: no need to use “-” in non-sense.
-
In Figure 1, A is not properly visible.
-
Line 179 and 180, please write the “IV-2 as mutation homozygous” and “IV-4 as wild-type homozygous” to remove any confusion.
-
Similarly Line 185, please write the “IV-2 and IV-5 as mutation homozygous affected” and “IV-4 as wild-type homozygous unaffected” to remove any confusion.
-
Line 217 Figure 4A, B change to Figure 5A and 5B
-
Line 219, please change Figure 4C to Figure 5C.
-
Line 226 and 227 correct the figure order.
-
Figure order in the manuscript is wrong after Figure 4. Author should correct order while writing in the manuscript.
-
In Table 3 and Table 5, Change the HEZ to HET as written in legend of Table HET: Heterozygous.

Reviewer 2 Report
The authors describe two unrelated Pakistani families that have postaxial polydactyly type A with intra- and inter-familial phenotypic variability. These families harbor the two different variants in KIAA0825 and GLI1 underlying the disease.
The article is well written and is interesting.
Some suggestions are:
- I suggest that table 1 be placed in only one page
- In tables 3 and 5, there is an error. The authors write “HEZ” instead of “HET”.
- In table 5, they cite “Palencia-Campo s et al., 2017”. Is it the same reference of table 6 “Palencia et al. 2017”? Please homogenize.
- In table 6, some cells do not have “-“ or “+”.
